# Exploring the Effects of LED-Based Visible Light Communication on Reading and Color Perception in Indoor Environments: An Experimental Study

**DOI:** 10.3390/s23062949

**Published:** 2023-03-08

**Authors:** Stefano Caputo, Lorenzo Mucchi, Regina Comparetto, Vittoria D’Antoni, Alessandro Farini, Valentina Orsi, Elisabetta Baldanzi

**Affiliations:** 1Department of Information Engineering, University of Florence, Via di S. Marta 3, 50139 Florence, Italy; stefano.caputo@unifi.it; 2CNR-INO, Istituto Nazionale di Ottica, Largo E. Fermi 6, 50125 Firenze, Italy

**Keywords:** visible light communications (VLC), human perceptions, psychophysical tests

## Abstract

Visible light communications (VLC) is a technology that enables the transmission of digital information with a light source. VLC is nowadays seen as a promising technology for indoor applications, helping WiFi to handle the spectrum crunch. Possible indoor applications range from Internet connection at home/office to multimedia content delivery in a museum. Despite the vast interest of researchers in both theoretical analysis and experimentation on VLC technology, no studies have been carried out on the human perceptions of objects illuminated by VLC-based lamps. It is important to define if a VLC lamp decreases the reading capability or modifies the color perception in order to make VLC a technology appropriate for everyday life use. This paper describes the results of psychophysical tests on humans to define if VLC lamps modify the perception of colors or the reading speed. The results of the reading speed test showed a 0.97 correlation coefficient between tests with and without VLC modulated light, leading us to conclude that there is no difference in the reading speed capability with and without VLC-modulated light. The results of the color perception test showed a Fisher exact test *p*-value of 0.2351, showing that the perception of color is not influenced by the presence of the VLC modulated light.

## 1. Introduction

The growth in mobile data traffic is driving the need for more spectrum, and the optical spectrum offers significant potential for increasing the available bandwidth. The optical spectrum has the potential to provide three orders of magnitude more capacity than the current RF spectrum. This can support the high data rate, low latency, and security requirements of future 6G wireless applications [1]. Visible light communication (VLC) is a promising optical wireless technology that uses the visible light spectrum for data transfer and localization [2,3]. It is based on the modulation of artificial light in the visible frequency range and has the potential to provide high data rates and low latency for various wireless applications [4].

Basically, any indoor lighting system, such as in a shopping center, office, or home, can potentially become the transmitter of a data communications system. Optical communications with visible light are also particularly suitable for applications such as indoor positioning systems (IPSs), to trace and find the position of a target in environments where GPS tracking is not possible, such as inside buildings [5,6]. A low cost solution for IPSs is based on an LED light as source and a smartphone camera as a receiver. This technology for the standard IEEE 802.15.7 [7] is called optical camera communication (OCC), and it appears only in a first revision of the standard. The definition of the range of frequencies to be used for OCC is postponed to the next revision of the standard. The image sensor of a smartphone typically consists of a number of pixels, each pixel contains a photodiode, which can be used as a VLC receiver [8]. This sensor is not designed to receive VLC signal and thus the performance is better for modulation at low frequencies.

The standard IEEE 1789–2015 [9] includes a definition of the concept of modulation frequencies for LEDs, and it also describes the health risks for humans about flickering light. This standard is followed especially by companies that produce LED drivers. The driver for dimming of LED light usually is achieved by a pulse width modulation (PWM) signal. The guidelines in the standard are limited to periodic signals, typically of LED drivers. The VLC modulation of LED light is not considered in this standard.

The approach of the standard IEEE 1789–2015 is basically a medical approach based on statistical occurrences of serious health symptoms, e.g., abnormal EEG responses, seizures from light stimuli, and epilepsy. In this document, we define, as a function of frequency and modulation depth, a critical, low-risk, and no observable effect level (NOEL) area.

The IEEE 802.15 Working Group completed a standard in 2011 for short-range wireless optical communication using visible light (IEEE Standard 802.15.7-2011). The Optical Wireless Communications (OWC) standard, which covers LED-ID, Optical Camera Communication (OCC), and LiFi, was revised in 2015 and included in the IEEE802.15.7r1 standard. The project is currently ongoing and aims to develop a standard that utilizes light wavelengths ranging from 10,000 nm to 190 nm in optically transparent media. In 2017, the group split, with 802.15.7m continuing to work on optical camera communications, while the IEEE 802.15.13 Task Group was established to work on multigigabit per second optical wireless communications, which uses high-speed photodiodes. The IEEE 802.15 Vehicular Assistant Technology Interest Group is also considering VLC as a communication option. In 2016, the IEEE 802.11 Working Group created a Topic Interest Group (TIG) to assess the potential technical and economic benefits of using light for wireless communications. The group received approval for their project authorization request in 2018, and the 802.11 Task Group bb is now responsible for developing the standard document. However, none of these groups is investigating if VLC-based light sources can alter human perceptions.

The International Telecommunications Union (ITU) produced a report in June 2018 (ITU-R SM.2422-0 “Visible light for broadband communications”), which discusses eye safety in the context of modulated light and the use of LED systems and visible light beams in visible light communication (VLC) and optical-beam-steered communication. It states that the retina is the most vulnerable part of the human eye and that the power exposure of visible light should be limited to avoid harm. The report also mentions that LED systems used for illumination purposes typically do not harm the retina, but the study recommends taking a closer look at safety issues for people who are working in close proximity to modulated light sources. Despite this recommendation, no publications are currently present in the literature on the potential alteration of human perceptions due to VLC-based light sources.

VLC systems have been proposed for various positioning applications in recent years. These systems use smartphone cameras to capture the light emitted by LED-based VLC beacons, which can be used to identify specific objects or to determine the location of a smartphone [10,11]. Some of these systems use triangular algorithms to calculate the position of a smartphone based on the lights of a single artwork [12], while others use an array of LEDs with different modulation to create a QR-code-like pattern that can be captured by the smartphone’s camera [13].

More recently, experiments on VLC have become quite popular. In [14], tests to evaluate the performances of discrete Fourier transform (DFT), discrete cosine transform (DCT), and discrete wavelet modulation (DWT) based orthogonal frequency division multiplexing (OFDM) schemes for VLC were reported. In [15], the results of experiments on VLC receivers containing two different sizes of commercially available silicon photomultipliers (SiPMs) are shown; in [16], current research on hybrid VLC and RF systems is reviewed.

However, the health risks to users are not described in these papers. The frequencies used in these studies are similar to those defined as critical in the IEEE 1789–2015 standard. It is important to note that more research is needed to understand the potential health risks associated with VLC systems and to ensure that they are safe for use. To the best of our knowledge, this topic has not been addressed in the literature.

To enable OCC technology to be used in everyday life, it is of importance to study the effects of information-modulated light sources on the perception of humans. In particular, it is important to investigate if the modulation of the light can affect the reading capability or the color perception of an individual. To the best of our knowledge, there are no studies on this topic published in the scientific literature.

The main contribution of this paper is the experimental evidence of a reasonable theoretical hypothesis: the use of VLC technology does not modify the perception of color or reading speed of human beings. This hypothesis was verified using some psycho-physical tests. For every test, the subjects performed the tests twice: the first time with a VLC-modulated light and the second time without VLC-modulated light, i.e., with a classical continuous light source. The results showed that visible light communications do not affect the color perception or reading capabilities of humans.

The remainder of this paper is organized as follow: The experimental setup and the psycho-physical tests used during the experimental campaigns are presented in Section 2 and Section 3, respectively. Section 4 reports the results of our experiments. Finally, our conclusions are summarized in Section 5.

## 2. Experimental Setup

A group of 20 volunteers (9 women and 11 men, all from the University of Florence and all of whom signed an informed consent), ranging in age from 25 to 60 years old, were administered reading tests (REX and Radner test) and color vision tests. All participants wore their prescribed eyeglasses or contact lenses, if necessary. The experiments took place in an office building room. A table and a chair were present in the room, and an LED lamp was the only light source present in the room. The lamp was set on the table, lighting the table where the tests were given (Figure 1).

The tests are detailed in the following sections, while the experiments are detailed and discussed in Section 4.

The OCC transceiver, used during experimental campaigns, was composed of a common marketed light bulb, typically used for desk lamps with a G4 socket and 12 V power supply, and an electronics driver to control the average voltage and modulation width. The block diagram of the system is shown in Figure 2, and each single block is described in the following.

In our implementation, the light signal at the output of the LED lamp was obtained by 3 blocks (Figure 2, left side): an Arduino block (which created the digital signal), a driver block (which created the electrical signal), and the LED light block (which converted the electrical signal to a light signal). The signal transmitted by the LED lamp can be represented by a time-dependent rectangular function
(1)$$s\left(t\right)=\sum_{j=0}^{N_{s}-1}s_{j}\left(t-jN_{s}T\right)$$
where
(2)$$s_{j}\left(t\right)=\sum_{k=0}^{N_{b}-1}[A_{1}\cdot \mathrm{rect}\left(\frac{t-T/2-kT}{T}\right)-A_{2}\cdot \mathrm{rect}\left(\frac{t-T/4-b_{k}T/2-kT}{T/2}\right)]$$
and
$$\mathrm{rect}\left(t/T\right)=\left\{\begin{array}{cc} 1 & \mathrm{if}\;\;-T/2<t<T/2\\ 0 & \mathrm{otherwise} \end{array}\right.$$

The variable $b_{k}\in \left\{0,1\right\}$ indicates the transmitted bit; parameter $A_{1}$ is the maximum amplitude of the rectangular function, while $A_{1}-A_{2}$ is the minimum. The maximum amplitude $A_{1}$ was set to 5 V in the Arduino digital signal, 3.3 V in the driver electrical signal, and 170 lumen in the light signal output (see Figure 2). Amplitude $A_{2}$ was used to set the minimum point of the rectangular function, and it was set to 5 V in the Arduino digital signal, 0.3 V in the driver electrical signal, and 17 lumen in the light signal output. The parameter *T* is the duration of the single rectangular function, $N_{b}$ is the number of bits in a single sequence, and $N_{s}$ is the number of transmitted sequences. In other words, a sequence of $N_{b}$ bits was created and then repeated $N_{s}$ times. The single bit sequence was selected as b={0,1,1,0}. The signal (Equation 2) was compliant with Manchester encoding [17]. The sequence b was selected to encompass all of the possible logic transitions appearing in Manchester encoding, where no more than two consecutive symbols can be of the same sign.

The lamp [18] was composed of 12 phosphor LEDs and a circuit regulating the current to keep the luminous intensity of the lamp as constant as possible. As an additional feature, by applying a voltage between 3 and 3.3 V across two pins of the current control circuit, normally unused, it was possible to dim the lamp from maximum emission all the way to a complete switch-off. This light bulb had a consumption of 2 W and a luminous flux of 170 lumen (similar to a 20 W traditional incandescent lamp). An Arduino board was used to control the voltage across these pins to convey digital information through the electronic driver.

Manchester encoding is generally used in VLC to keep the average light intensity constant for any sequence of transmitted bits [19]. In addition, this encoding does not allow the transmission of more than two consecutive equal symbols in order to avoid the flickering caused, for example, by a series of consecutive identical symbols.

During the measurements, the luminous intensity was set to 95% of the standard intensity of the lamp, the amplitude modulation to 10% to achieve a lamp emission oscillating between 90% and 100%, the intensity subcarrier frequency was set to 180 Hz, and the frequency of modulation was set to 90 Hz. The modulation frequency is in a critical area according to standard IEEE 1789–2015, while the intensity subcarrier frequency is in a NOEL area, which means that the perception of flicker is minimal. These frequencies were selected after preliminary tests, in which the perception of flickering was evaluated.

## 3. Psycho-Physical Tests Used during Experimental Campaign

### 3.1. Reading Tests

The Reading Explorer (REX) and Radner tests can be useful in evaluating the potential effects of VLC technology on reading activities. The REX test can provide information on how VLC technology affects reading speed, comprehension, and accuracy, while the Radner test can provide information on how VLC technology affects visual acuity.

#### 3.1.1. Reading Explorer Test (REX Test)

The REX test (Figure 3a) is an eye chart that allows estimating the reading performance while changing the text/background contrast levels [20]. The REX test presents two sets, each one with four charts. Every page of the test has three different phrases arranged on three lines with aligned margins. Each phrase has 60 letters. The contrast of each phrase decreases with a logarithmic progression from top to bottom. The subject respects a reading distance of 40 cm.

On the REX charts, each letter is equivalent to 1.0 logMAR in size. The first line of the REX charts has a text/background contrast of 89.13%, which can be detected with a minimum contrast sensitivity of 1.122%, or 0.05 in logarithmic units. The contrast sensitivity required to read each subsequent line increases by 0.15 logarithmic units from the previous line, up to a maximum of 1.7 units for the 12th and final line [20]. The subjects read the phrases aloud, starting from the one with the higher contrast. The following phrases are covered. The person who conducts the test annotates the nonread words or the one read incorrectly. The subject is invited to read as quickly yet precisely as possible. The reading time in seconds for each phrase $t_{REX}$ (s) is measured to calculate the reading speed, $v_{REX}$.
(3)$$v_{REX}=\frac{600}{t_{REX}}$$

#### 3.1.2. Radner Reading Test

The Radner Reading test (Figure 3b) [21] is an eye chart that allows evaluating the near visual acuity. The chart presents 24 phrases that present a standardized structure. From top to bottom, the font size decreases by 0.1 logarithmic units. The reading time $t_{Radner}$ (s) was measured, and the reading speed $v_{Radner}$ was calculated using the following equation [22]:(4)$$v_{Radner}=\frac{14\times 60}{t_{Radner}}$$

### 3.2. Color Vision Tests

The *Ishihara test*, the *Farnsworth–Munsell 100 Hue Colour Vision*, and the *City University Colour Vision tests* can provide information on how VLC technology affects color perception.

#### 3.2.1. Ishihara Color Test

The Ishihara test (Figure 4a) is a pseudoisochromatic plate test [23]. It consists of 38 plates; each of the plates presents dots with different sizes and colors but with the same luminosity. Some of these dots form a path or a number easily distinguishable by a person with normal color perception. If the examined subject experiences difficulty or is unable to determine the number or path of the plates, the subject is identified as having anomalous color sensitivity. The first plate (with number 12) is not a pseudoisochromatic image; thus, the number is also visible for a subject with altered color perception. Errors in the identification of the figure on plates 2 to 17 represent anomalies in the perception on the R/G axis. Plates 18 to 21 are visible only those who have an altered perception of the colors on the R/G axis. Plates 22 to 25 allow distinguishing color blindness (protanopia for red; deuteranopia for green) from partial color blindness (protanomaly or deuteranomaly). Plates 27 to 38 present paths that the subject can indicate and allow the administration of the test to illiterate subjects. The subject needs to stay at a distance of 30–40 cm, using right optical correction if needed.

#### 3.2.2. City University Colour Vision Test

The City University Colour Vision Test (CUT) [24] is a matching test (Figure 4b). It consists of two parts: with the first one, it is possible to detect color vision deficits; the second part of the test allows the evaluation of the severity of the anomaly. With the CUT, it is possible to identify subjects with tritanopia (color blindness toward the blue wavelengths). Initially, the tests present four columns of colored dots, two in the upper part and two in the lower part of the page. The task of the patient is to identify the dots that are different in color from the other ones in the same column. A subject with normal color sensitivity should be able to identify correctly 9–10 of the dots. The second part of the test has 10 tables; each table presents a central dot with four other dots around it. The patient must indicate which out of the four peripheral dots is the most similar in color to the central one.

#### 3.2.3. Farnsworth–Munsell 100 Hue Colour Vision Test

The Farnsworth–Munsell 100 Hue Colour Vision test (Figure 4c) is an arrangement test; the subject is asked to put the shown colored plates in the right order [25]. If the subject finds the assigned task difficult to accomplish, it indicates an anomaly in their color vision [26]. The Farnsworth–Munsell 100 Hue Colour Vision test consists in 85 colored plates, numbered from 1 to 85, divided into 4 groups (the first set contains the colors from 85 to 21, the second from 22 to 42, the third from 43 to 63, and the fourth from 64 to 84). The examined subject is given each test one at a time and orders the plates of the set. The examiner checks the order of the number signed on the back of each colored plate and writes down the results. Once all the four groups have been ordered, it is possible to calculate the error score of every single color (Equation (Equation 5)).
(5)ErrorScore=|n−(n−1)|+|n−(n+1)|
where *n* is the number indicating a certain color.

If the colors are put in the right order, then ErrorScore=2; thus, 170 is the best score a subject can obtain. The total error score (*TES*) is calculated by summing each single error score and then deducting it from 170. As the number of errors increases, the total score increases because, when a plate is misplaced, the ErrorScore>2. Verriest et al. [27] showed that the distribution of the total error score is not Gaussian and that the square root of the total error score has a Gaussian distribution. The Farnsworth–Munsell100 Hue Colour Vision test demands good collaboration from the examined subject. It takes about 20 min to perform the test. The results can be affected by the age of the subject, visual acuity, and retinal illumination.

## 4. Experimental Results

### 4.1. Reading Tests

A group of 20 subjects, consisting of 9 women and 11 men with ages ranging from 25 to 60 years, participated in reading tests (REX and Radner tests). Every subject wore optical correction if needed.

The subjects had to read the twelve phrases of the REX test, one time with the VLC lamp ON and another time with the lamp turned OFF without them knowing it. Reading times ($t_{ON}$, $t_{OFF}$) were measured for each phrase with a different contrast. Thus, the reading speeds ($v_{ON}$, $v_{OFF}$) were calculated using the Equation (Equation 3).

For each subject, the critical logarithmic contrast sensitivity was identified, which corresponds to the minimum level of contrast that allowed the subject to read the phrase with the same speed that the subject reached in the first black-on-white (maximum contrast) phrase. In Figure 5, it can be noted that after that after a critical value, the reading speed abruptly decreases.

The average reading speed was calculated for each subject, after excluding the reading speed values past the critical point. As a result, the averages and standard deviations, ${\bar{v}}_{ON}$ and $\sigma_{ON}$, ${\bar{v}}_{OFF}$ and $\sigma_{OFF}$, respectively, were obtained for every subject.

The average reading speeds and the corresponding statistical errors for each subject are presented in Figure 6.

The values for reading speed were in a reasonable range for adults [28]. The axis of the ordinates reports the ${\bar{v}}_{ON}$, and the horizontal bars represent $\sigma_{ON}$. The axis of the abscissas reports the ${\bar{v}}_{OFF}$, and the horizontal bars represent $\sigma_{OFF}$. The figure also illustrates the equation of the linear fit and the Pearson coefficient R. By observing Figure 6, it can be noted that there is a strong positive correlation between the reading speed measured when the device was on and when it was off. Therefore, we concluded that the presence of the VLC lamp did not influence the contrast sensitivity of the patients.

The same subjects were asked to participate in the Radner test. In a similar manner as the REX test, the reading times for phrases corresponding to various visual acuities (*VA*) were measured. The reading speeds ($v_{ON}$, $v_{OFF}$) were calculated using Equation (Equation 4). Similar to the REX test, the critical values of VA were detected for each participant, and the average and standard deviation of the reading speeds were determined, after excluding the reading speed values past the critical point. In Figure 7, the values of ${\bar{v}}_{ON}\pm \sigma_{ON}$ and ${\bar{v}}_{OFF}\pm \sigma_{OFF}$ of the subjects obtained from the Radner test are reported. Similar to the REX test, no significant change in reading speed with VLC ON and OFF was observed in the Radner test (R=0.91). Therefore, also for the Radner test, we concluded that the presence of the VLC lamp did not influence the contrast sensitivity of the patients or their reading times.

A Bland–Altman plot [29] was also drawn for the results of both the REX (Figure 8) and Radner tests (Figure 9). The statistical analysis for the REX test resulted in a bias of 0.3500 and a standard deviation of the difference between the two measurements of 7.6590. For the Radner test, the bias was 0.3000 and the standard deviation of the difference was 12.7325.

### 4.2. Colour Vision Tests

A group of 20 subjects, consisting of 11 women and 9 men with ages ranging from 20 to 65 years, also participated in the color vision test. All of the subjects wore optical correction, if needed.

For the Ishihara test and the CUT, the number of errors made by the subjects were counted when the VLC lamp was on and when it was off. Figure 10 reports the registered errors with the Ishihara test. To understand if the differences were statistically significant, the *Fisher exact test* [30] was conducted. The *p*-value was 0.2351, which is much greater than 0.05, which showed that the reading speed (of colored texture) was not influenced by the presence of the VLC device. As such, it was not necessary to conduct any statistical tests for the CUT, as the same number of errors was observed with the VLC on and off.

As shown in Figure 11, the square roots of the total error scores made by the 20 subjects in the Farnsworth–Munsell 100 Hue Colour Vision Test were compared when the VLC lamp was on and off, with age-tabulated values of mean and statistical errors of $\sqrt{TES}$ [27]. It can be observed that all the subject had rather similar $\sqrt{TES_{OFF}}$ and $\sqrt{TES_{OFF}}$, with the exception of subjects 11 and 14, who displayed the highest differences between the two values ($|\Delta_{max}|=2.59$ for subject 11). Moreover, 15 subjects over 20 were found to be within the values acceptable for their age range [27], with the exceptions of subject 3, 5, 8, 12, and 14. Subject 3 had $\sqrt{TES_{OFF}}$ and $\sqrt{TES_{OFF}}$ values that were much higher than the average of their peers; this indicated that this patient may have some difficulties in the color discrimination, even if it was not detected by the other color vision tests (this could be an hint toward a tritan anomaly). Subject 5 had a better $\sqrt{TES_{OFF}}$ than those in their age range and a $\sqrt{TES_{OFF}}$ at the lower limit of this interval. Subject 8 had $\sqrt{TES_{OFF}}$ and $\sqrt{TES_{OFF}}$ values lower than the average of their peers, showing better capabilities in the color discrimination task. Subject 12 presented a higher $\sqrt{TES_{OFF}}$, and subject 14 had a $\sqrt{TES_{OFF}}$ slightly below the average.

To summarize the results regarding the Farnsworth–Munsell 100 Hue Colour Vision test a *t*-test was conducted comparing the $\sqrt{TES}$ with VLC OFF and with VLC ON. There was no significant difference in the two $\sqrt{TES}$t(19)=−0.97, p=0.342. In Figure 12, there is a dispersion plot of the two $\sqrt{TES}$. The best fit for our data using the least-squares method without intercept reported an angular coefficient α=1.02±0.04, well compatible with $\sqrt{TES_{OFF}}=\sqrt{TES_{ON}}$. The correlation coefficient was $R^{2}=0.75$.

## 5. Conclusions

In this study, we aimed to demonstrate that visible light communication (VLC) technology does not negatively impact human perceptions. Because VLC is envisioned to be a pervasive technology in the future (see, e.g., current 6G vision), it is important to investigate if information-modulated light sources affect the human capability to perceive color or read text.

A significant experimental campaign was carried out by submitting color and reading tests to volunteers, where each test was first lit with VLC on (modulated light) and then with VLC off (nonmodulated light).

Our experiments showed that there were no differences in reading speed with or without data transmission through the light source. The tests (REX and Radner) showed an almost perfect linearity of the regression curve, with a Pearson correlation coefficient of R=0.97 and R=0.91, respectively, between volunteer reading speed with and without VLC-modulated light. The statistical analysis for the REX test resulted in a bias of 0.3500 and a standard deviation of the difference between the two measurements of 7.6590. For the Radner test, the bias was 0.3000, and the standard deviation of the difference was 12.7325. This finding can assure that VLC technology can be used in offices, schools, or other locations where reading plays an important role.

Our experiment also demonstrated that the human color perception and discrimination were not affected by the VLC-based light. This suggests that VLC technology may also be useful in environments (or types of work) where color perception is important, such as museums, in the textile industry, etc. The Fisher exact test, which was applied to the results of the Ishihara test, showed a *p*-value equal to 0.2351, which is much greater than 0.05, showing that the perception of color was not influenced by the presence of the VLC-modulated light.

In future work, the effect of VLC with long-term testing (over a few hours or an entire day) can be investigated. Additionally, these results can be validated with other methodologies such as the use of electroencephalography (EEG) sensors to determine safe frequencies for the human eye that can be perceived by a sensor, as suggested in [31].

## Figures and Tables

**Figure 1 sensors-23-02949-f001:**
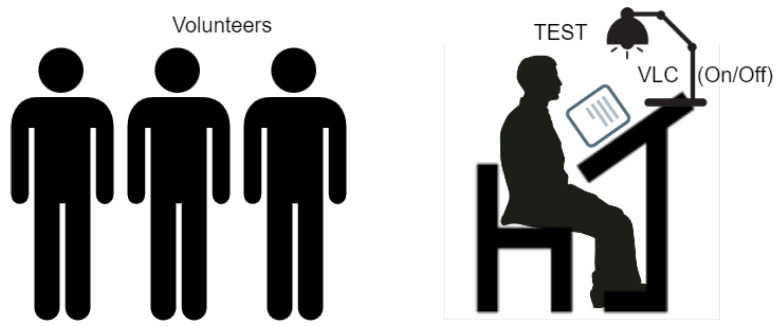
Sketch of the test set up.

**Figure 2 sensors-23-02949-f002:**
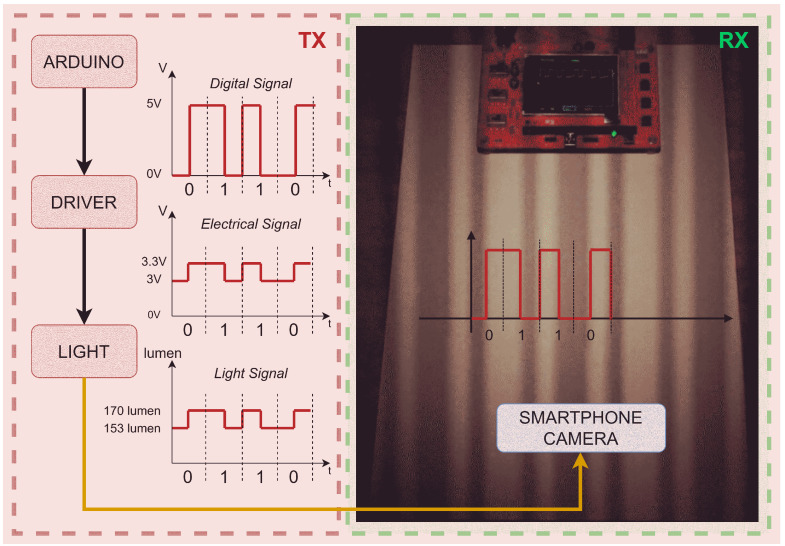
Block diagram of the transmission system used for the experiments. On the left, the block diagram of the transmitting chain; on the right, a photograph to demonstrate the data transmission into the light.

**Figure 3 sensors-23-02949-f003:**
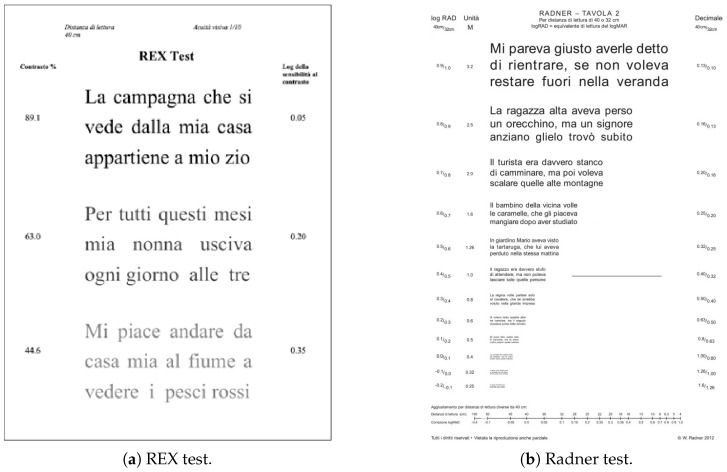
The reading tests.

**Figure 4 sensors-23-02949-f004:**
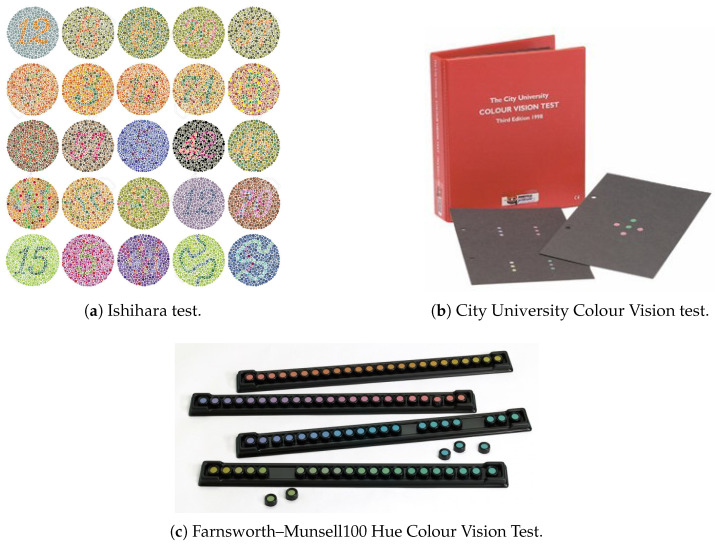
Color vision tests.

**Figure 5 sensors-23-02949-f005:**
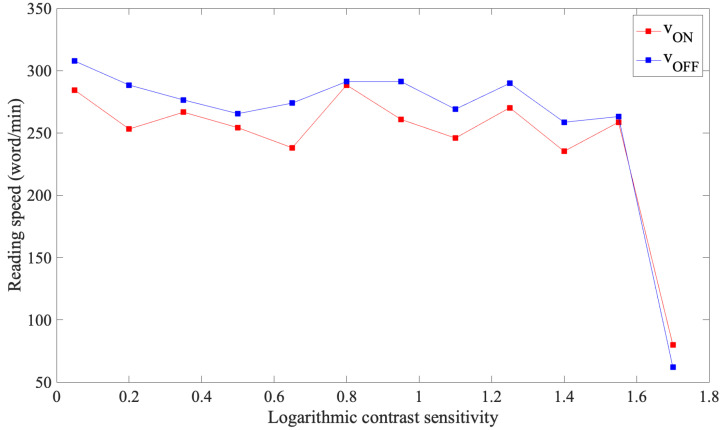
Reading speed depending on the logarithmic contrast sensitivity of subject 1 with the REX test. Observing the performance of subject 1, when the logarithmic contrast sensitivity is equal to 1.55, the reading speed values abruptly decrease. When this occurs, the critical contrast sensitivity value has been reached, beyond which the subject is unable to successfully complete the task.

**Figure 6 sensors-23-02949-f006:**
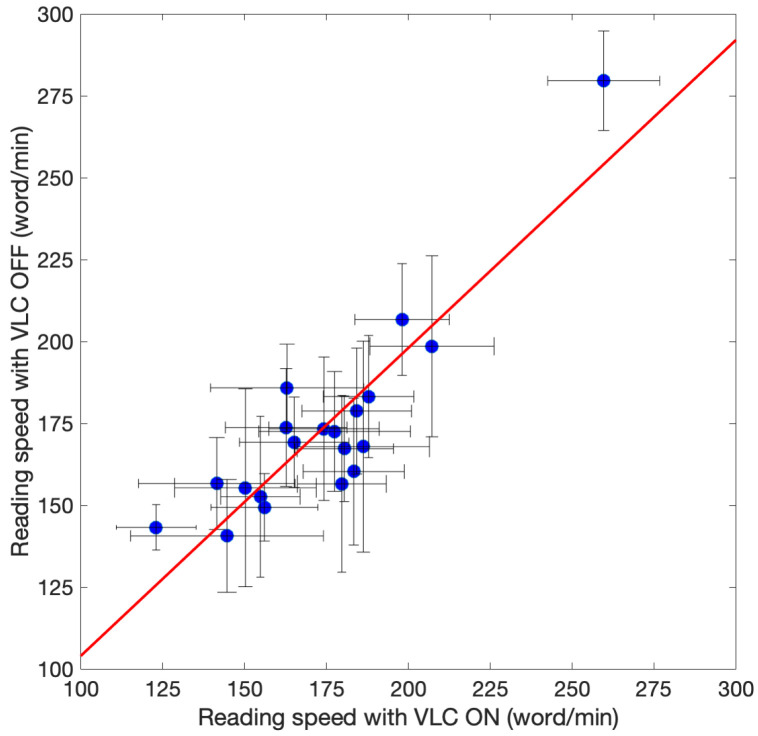
Scatter plot between ${\bar{v}}_{ON}\pm \sigma_{ON}$ and ${\bar{v}}_{OFF}\pm \sigma_{OFF}$ for each examined subject when taking the REX test. The figure also displays the linear regression fit, represented by the equation y=1.02x−4.60, and the Pearson correlation coefficient R=0.97. It can be noted that there is a positive correlation between ${\bar{v}}_{ON}$ and ${\bar{v}}_{OFF}$.

**Figure 7 sensors-23-02949-f007:**
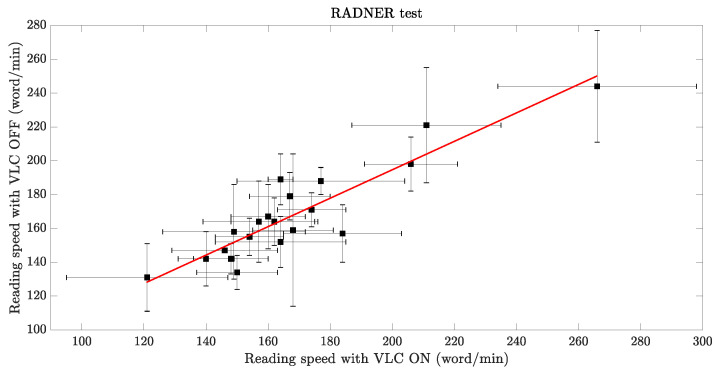
Scatter plot between ${\bar{v}}_{ON}\pm \sigma_{ON}$ and ${\bar{v}}_{OFF}\pm \sigma_{OFF}$ for each examined subject when taking the Radner test. The figure also displays the linear regression fit, represented by the equation y=0.84x+26.53, and the Pearson correlation coefficient R=0.91. There is a positive correlation between ${\bar{v}}_{ON}$ and ${\bar{v}}_{OFF}$.

**Figure 8 sensors-23-02949-f008:**
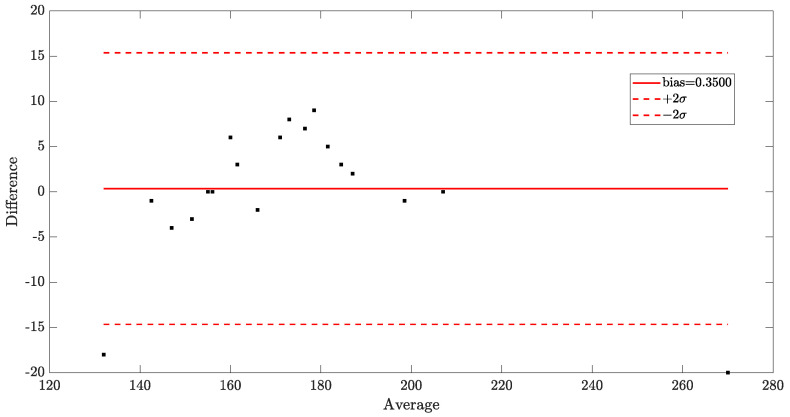
Statistical results of the REX test. The differences in subject performance with or without transmission data. The bias was 0.3500, and the standard deviation of the difference was 7.6590, as defined in [29].

**Figure 9 sensors-23-02949-f009:**
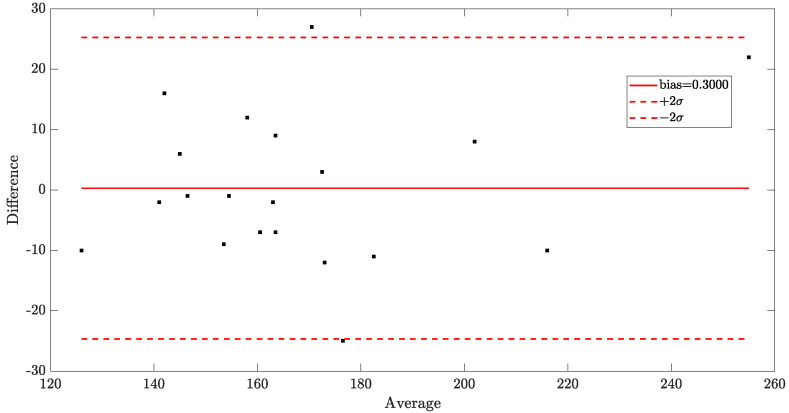
Statistical results of the Radner test. The differences in subject performance with or without transmission data. The bias was equal to 0.3000, and standard deviation of the difference was 12.7325, as defined in [29].

**Figure 10 sensors-23-02949-f010:**
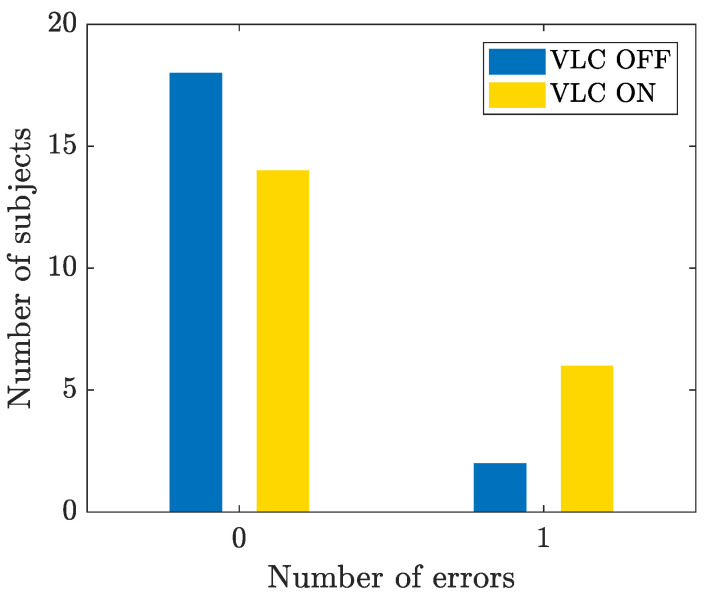
Bar plot representing the number of errors for the Ishihara test with VLC on (yellow bar) and off (blue bar).

**Figure 11 sensors-23-02949-f011:**
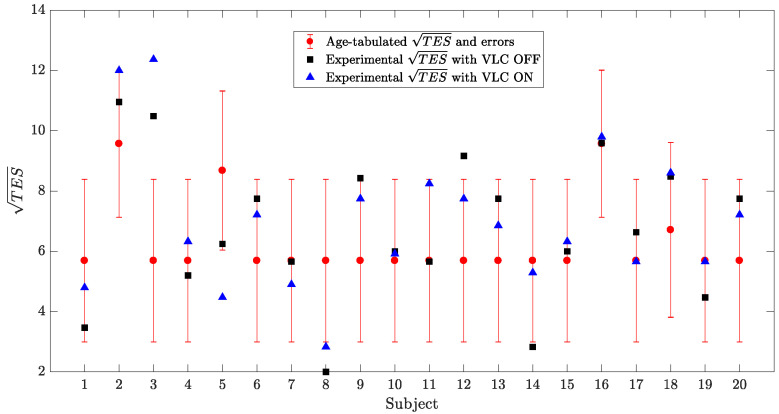
Representation of square roots of the total error score values for each subject. In red, the average values of $\sqrt{TES}$ and their statistical error based on the age range of the patient are reported [27]; in blackm the $\sqrt{TES}$ measured with VLC OFF is reported; and in blue, the $\sqrt{TES}$ measured with VLC ON is reported.

**Figure 12 sensors-23-02949-f012:**
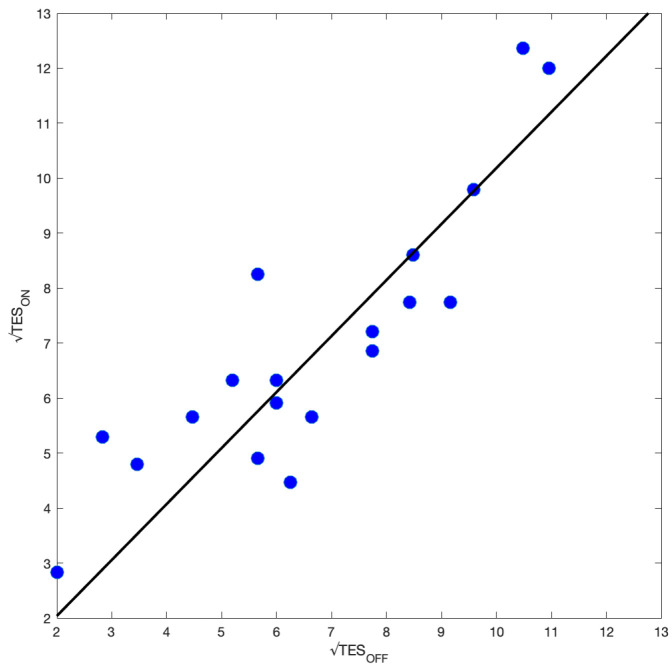
$\sqrt{TES_{ON}}$ versus $\sqrt{TES_{OFF}}$. The black line is the best fit made using the least-squares method without intercept.

## Data Availability

Not applicable.

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
