# Peer review of "Exploring the Effects of LED-Based Visible Light Communication on Reading and Color Perception in Indoor Environments: An Experimental Study"

_sensors, 2023, doi:10.3390/s23062949_

Round 1

Reviewer 1 Report

Technical Soundness

  1. The proposed system model for experimental set up is not clear for both transceivers and the overall presentation of the paper is not connected with the step-wise equation. 
  2. Detailed mathematical analysis of VLC communication access capability is not clear.
  3. How to generate and be compatible with the transmission bits by adopting different modulation and complexity analyses of the proposed work is not clear.
  4. The compared work is not provided with references.  
  5. A detailed system model is required with each component specification.
  6. The VLC channel model is not explained elaborately.
  7. Plagiarism is 25%, it should be less than 18%.

         Presentation of Paper

1.     Readability of the figures needs to be improved. Figure 4 and Figure 5 need to revise

2.     Detailed mathematical analysis is required for clear vision of proposed work.

3.     A detail comparison of results is required with the existing work for better understanding and validating of paper.

Author Response

Reviewer #1

Reply to Reviewer #1: We would like to thank the Reviewer very much for the constructive comments and suggestions, which gave us the possibility to improve the overall quality of the manuscript. A point-by-point answer to the Reviewers’ comments is reported below. 

Comment 1. The proposed system model for experimental set up is not clear for both transceivers and the overall presentation of the paper is not connected with the step-wise equation. 

Reply to Comment 1. Thank you for this significant suggestion. We have improved the description of the experimental set up. We have also added a figure as an illustration of the experimental set up in order to give a quicker insight of it. We have also added a paragraph with all the equations showing how the transmitted signal is generated. Anyway, it is important to point out that the paper is basically related to the reading and colour perception tests, and not to the VLC system used to modulate the light of the lamp which has been used to enlighten the tests. 

Comment 2. Detailed mathematical analysis of VLC communication access capability is not clear. 

Reply to Comment 2. Thank you for this comment, it is important for us to better point out the real objectives of our article. Actually, the proposed paper does not deal with technical details on VLC communication links. The paper aims to demonstrate if reading and colour perception capabilities of human beings suffer the effects of a VLC modulated light. To the best of the authors' knowledge, this aspect is not investigated in literature. 

Said this, we additionally want to stress out that there is no “access”, since there is no sharing of the communication medium. The VLC channel is covered only by a square wave signal as depicted in Fig. 2 of the manuscript. An Arduino board is used to generate a voltage square wave with Manchester encoding. This signal is then fed to a Driver board which converts the voltage signal from 0-5 V to 3-3.3 V. The last signal is then used to pilot the lamp so that the light ranges between 153 and 170 lumen. 

We have inserted the mathematical model of the transmitted signal in the revised version of the manuscript. 

Comment 3. How to generate and be compatible with the transmission bits by adopting different modulation and complexity analyses of the proposed work is not clear.

Reply to Comment 3. Thank you for pointing out this important issue. In our scheme the VLC system is extremely simple: just a square wave with Manchester encoding to keep the average level of light intensity at the correct level (the average lumen must be the same, with and without VLC). Although we did not implement complex modulations in our system since it was not the scope of the paper, different modulation schemes can be adopted by the VLC system, some of those are simple others are more complex. The most common VLC modulation schemes are:

  1. On-Off Keying (OOK): This is the simplest VLC modulation scheme, where the data is encoded as the presence or absence of light. The absence of light represents a binary 0, and the presence of light represents a binary 1.
  2. Pulse Amplitude Modulation (PAM): This is a more complex VLC modulation scheme that encodes data by varying the intensity of the light pulses. PAM can achieve higher data rates than OOK.
  3. Frequency Shift Keying (FSK): This VLC modulation scheme encodes data by varying the frequency of the light signal. The data is represented by different frequency shifts, such as a high frequency for a binary 1 and a low frequency for a binary 0.
  4. Color Shift Keying (CSK): This VLC modulation scheme uses different colours of light to represent different data symbols. For example, red light can represent a binary 1, while green light can represent a binary 0.
  5. Orthogonal Frequency Division Multiplexing (OFDM): This is a more advanced VLC modulation scheme that uses multiple sub-carriers to encode data. OFDM can achieve high data rates and is also robust to interference.

Each of these VLC modulation schemes has its own advantages and disadvantages, and the choice of modulation scheme depends on the specific application requirements.

In general, the complexity of a VLC system depends on several factors, such as the modulation scheme used, the type of receiver and transmitter, and the desired data rate and range. In general, VLC systems are less complex than some other wireless communication systems, such as radio frequency (RF) systems, because visible light signals do not penetrate walls and can be easily confined to a room or a specific area. This simplifies the design of the system, as there is less need for complex signal processing and interference mitigation techniques.

However, there are some challenges in designing a VLC system that can affect its complexity. For example, visible light signals can be affected by shadows, reflections, and interference from other light sources, which can impact the reliability and range of the system. To mitigate these effects, VLC systems may require complex optical components such as lenses, filters, and beam-steering devices to direct the light signals and reduce interference.

In addition, the complexity of the VLC system can increase with the desired data rate, as higher data rates require faster modulation and demodulation techniques and more precise synchronisation between the transmitter and receiver. This can increase the computational requirements of the system and require more powerful processing units.

In case of OCC, the typical modulation used is OOK, as in the IEEE802.15.7r1 standard. The VLC system proposed in this paper does not need to be so complex since the transmission is intended from a desk lamp to the desk table. Our system in fact uses OOK modulation compliant to the IEEE802.15.7r1 standard. This work is intended as a first insight into the investigation of the influence of VLC modulated light on the reading and colour perception of human beings. 

Comment 4. The compared work is not provided with references.  

Reply to Comment 4. Authors would like to thank the reviewer for the comment, but we were not able to completely understand what the reviewer intended to suggest. There is no a “compared work” referred in the paper. If the reviewer intended to refer to other similar papers in literature, to the best of our knowledge there is no other paper dealing with the influence of VLC on human reading and colour perception. 

Comment 5. A detailed system model is required with each component specification.

Reply to Comment 5. Thank you for this comment. Actually, we are not sure to have completely understood what the reviewer wanted to suggest. If the reviewer intended to request a description of each block that composes the VLC system which enlightens the reading and colour perception tests, this description is already present in the paper (see Sec. 2). 

Let us additionally state that the proposed paper does not deal with technical details on VLC communication links. The paper aims to demonstrate if reading and colour perception capabilities of human beings suffer the effects of a VLC modulated light. 

The VLC channel is covered only by a square wave signal as depicted in Fig. 2 of the manuscript. An Arduino board is used to generate a voltage square wave with Manchester encoding. This signal is then fed to a Driver board which converts the voltage signal from 0-5 V to 3-3.3 V. The last signal is then used to pilot the lamp so that the light ranges between 153 and 170 lumen. 

We have inserted the mathematical model of the transmitted signal in the revised version of the manuscript. 

Comment 6. The VLC channel model is not explained elaborately.

Reply to Comment 6. Thank you for this comment. As stated in the reply to comment 2, the proposed paper does not deal with technical details on VLC communication links, including the channel modelling. The paper aims to demonstrate if reading and colour perception capabilities of human beings suffer the effects of a VLC modulated light. Anyway, we agree with the reviewer that the experimental set up was not described properly. We have improved the description of the experiments in the revised version of the manuscript.     

Comment 7. Plagiarism is 25%, it should be less than 18%.

Reply to Comment 7. Thank you very much for pointing out this important issue. Actually, if you remove the bibliography, the similarity score becomes 12% (by using Turnitin software). Anyway, in the revised version of the manuscript we have removed two larger parts of text which contributed to increase the similarity score. 

Comment 8. Readability of the figures needs to be improved. Figure 4 and Figure 5 need to revise 

Reply to Comment 8. Thank you for highlighting this issue. We have improved the quality of  Figures 4 (now figure 5) and 5 (now figure 6) as well as the readability of the text over the axes and legenda. Now in figure 6 (figure 5 in the first version) we have also changed the ratio of the axes, to better show the relationship between the two variables.

Comment 9. Detailed mathematical analysis is required for clear vision of proposed work.

Reply to Comment 9. Thank you for this comment. As stated in the reply to comment 2, the proposed paper does not deal with technical details on VLC communication links. The paper aims to demonstrate if reading and colour perception capabilities of human beings suffer the effects of a VLC modulated light.

Comment 10. A detail comparison of results is required with the existing work for better understanding and validating of paper.

Reply to Comment 10. Thank you for your important comment. We agree with the reviewer that a comparison with other existing works would be an added value for our paper, but to the best of authors’ knowledge there are no other papers on this topic in literature at the moment. 

Reviewer 2 Report

Thank you for this interesting work and the new idea.

# The main question addressed by the research is the possibility of  psychophysical tests on humans to define if the VLC lamp modifies the perception of colours or the reading speed.

# I see the topic relevant in the field. It address a specific gap in the field. (New idea).

# The subject is familiar, but, the novelty (in my opinion) is the use of VLC system to achieve it. Unfortunately, I do not see a comparison (of obtained results) with a previously published work.

# The conclusions are consistent with the evidence and arguments presented and they address the main question posed, but there is a need for some numerical values.

------------------------------------------------

I have some comments that must be considered in the modified manuscript.

---------------------------------------------------------------------------

1) Both (Abstract) and (Conclusion) are descriptive. Both need some numerical values of the main findings.

2) Eq. (2) needs a reference.

3) Reading speed in Figs. 4=5 is NOT compared with a previous work. or: at least tell me how can you judge the correctness of your results? Are the values (results) in the reasonable range? (I do NOT know).

4) Please, focus on the importance of Fig. 6 and write more discussion.

5) Discussion of Fig. 10 does not fulfill the correctness of the method used. It only discusses the case of patient! In contrast to this, the discussion of Fig. 9 gives a meaning for the results and what is better and why.

6) I see 5 recent references (two in 2022 and 3 in 2021) out of 25 references is too small. Can you please use more recent references?

Author Response

Reviewer #2

Thank you for this interesting work and the new idea.

# The main question addressed by the research is the possibility of  psychophysical tests on humans to define if the VLC lamp modifies the perception of colours or the reading speed.

# I see the topic relevant in the field. It address a specific gap in the field. (New idea).

Reply to Reviewer #2: We would like to thank the Reviewer very much for the interest expressed for our paper and for the constructive comments and suggestions, which gave us the possibility to improve the overall quality of the manuscript. A point-by-point answer to the Reviewers’ comments is reported below. 

Comment 1. The subject is familiar, but, the novelty (in my opinion) is the use of VLC system to achieve it. Unfortunately, I do not see a comparison (of obtained results) with a previously published work.

Reply to Comment 1. Thank you for your important comment. We agree with the reviewer that a comparison with other existing works would be an added value for our paper, but to the best of authors’ knowledge there are no other papers on this topic in literature at the moment. 

Comment 2. The conclusions are consistent with the evidence and arguments presented and they address the main question posed, but there is a need for some numerical values.

Reply to Comment 2. Thank you for this suggestion. We have improved the Conclusion section in the revised version of the manuscript. 

Comment 3. Both (Abstract) and (Conclusion) are descriptive. Both need some numerical values of the main findings.

Reply to Comment 3. Thank you for pointing out these important points. We have improved both the Abstract and the Conclusion sections in the revised version of the manuscript, according to the reviewer’s suggestion. 

Comment 4. Eq. (2) needs a reference.

Reply to Comment 4. We have added a reference (Radner et al. 2014) where this formula is used (it is presented as a phrase in that paper, not as a formula “(14 words×60 s divided by the reading time)”. 

Comment 5. Reading speed in Figs. 4=5 is NOT compared with a previous work. or: at least tell me how can you judge the correctness of your results? Are the values (results) in the reasonable range? (I do NOT know).

Reply to Comment 5. Thank you for this comment. We have added a sentence explaining that such results are in a reasonable range and we have added a reference to a metanalysis with a lot of data. 

Comment 6. Please, focus on the importance of Fig. 6 and write more discussion.

Reply to Comment 6. Thank you for this comment. We have added a sentence about this result, explaining that the Radner test also confirms that VLC has no effect on visual skills. 

Comment 7. Discussion of Fig. 10 does not fulfill the correctness of the method used. It only discusses the case of patient! In contrast to this, the discussion of Fig. 9 gives a meaning for the results and what is better and why.

Reply to Comment Thanks for this comment. Indeed, the reviewer is right, there is a sort of lack of statistical analysis in the first version of the paper. We have improved the statistical analysis by adding to the new version of the paper the t-test, a new figure and a least squares method showing that there is no difference between the two situations. 

Comment 8. I see 5 recent references (two in 2022 and 3 in 2021) out of 25 references is too small. Can you please use more recent references?

Reply to Comment 8. Thank you for this important suggestion. We have inserted (and discussed) more recent references in the revised version of the manuscript. 

Reviewer 3 Report

In this work, authors have demonstrated the results of psychophysical tests on humans to define if the VLC lamp modifies the perception of colours or  the reading speed. The manuscript is written in good style.  The overall structure of the manuscript is also very good. I have just few minor comments which the authors can consider as suggestions to improve further the quality of this work. Please check my comments as below: 

1) Introduction section can be improved. The authors need to add some recent works on recent innovations in VLC technology specially related to the dimming part.  Some of the works are mentioned below: 

a) https://doi.org/10.1117/12.2522838

b) 10.1109/JPHOT.2023.3239112

c) 10.1109/MCOM.001.2100748

2) The conclusion section needs to elaborate further to include the findings from the different tests. 

Author Response

Reviewer #3

In this work, authors have demonstrated the results of psychophysical tests on humans to define if the VLC lamp modifies the perception of colours or  the reading speed. The manuscript is written in good style.  The overall structure of the manuscript is also very good. I have just few minor comments which the authors can consider as suggestions to improve further the quality of this work. Please check my comments as below: 

Reply to Reviewer #3: We would like to thank the Reviewer for the appreciation expressed about our work and for the constructive comments and suggestions, which gave us the possibility to improve the overall quality of the manuscript. A point-by-point answer to the Reviewers’ comments is reported below. 

Comment 1. Introduction section can be improved. The authors need to add some recent works on recent innovations in VLC technology specially related to the dimming part.  Some of the works are mentioned below: 

  1. a) https://doi.org/10.1117/12.2522838
  2. b) 10.1109/JPHOT.2023.3239112
  3. c) 10.1109/MCOM.001.2100748

Reply to Comment 1. Thank you very much for suggesting these articles. They have been all inserted in the revised version of the manuscript and discussed in the introduction section. 

Comment 2. The conclusion section needs to elaborate further to include the findings from the different tests. 

Reply to Comment 2. Thank you for the significant suggestion. We have improved the conclusions section in the revised version of the manuscript. 

Round 2

Reviewer 1 Report

Authors went through the comments and upadated the article in a better shape with clarity. I am happy to recommend this article for the publication.